Awareness and knowledge of Chikungunya infection following its outbreak in Pakistan among health care students and professionals: a nationwide survey

Mallhi Tauqeer Hussain tauqeer.hussain.mallhi@hotmail.com 1
Khan Yusra Habib yusrahabib@ymail.com 2
Tanveer Nida 3
Bukhsh Allah 4
Khan Amer Hayat 5
Aftab Raja Ahsan 6
Khan Omaid Hayat 7
Khan Tahir Mehmood 4
1 Department of Pharmacy Practice, Faculty of Pharmaceutical Sciences, Government College University Faisalabad , Faisalabad , Pakistan
2 Institute of Pharmacy, Lahore College for Women University , Lahore , Pakistan
3 Punjab Medical College , Faisalabad , Pakistan
4 Institute of Pharmaceutical Sciences, University of Veterinary and Animal Sciences , Lahore , Pakistan
5 Discipline of Clinical Pharmacy, School of Pharmaceutical Sciences, Universiti Sains Malaysia , Malaysia
6 School of Pharmacy, Taylor’s University , Subang Jaya , Selangor , Malaysia
7 Department of Pharmacy, The University of Lahore , Lahore , Pakistan
Wiles Siouxsie
Electronic publication date: 2018 Aug 30
Publication date: 2018
Volume: 6
Electronic Location ID: e5481
Received 2018 Mar 9; Accepted 2018 Jul 30
Copyright: ©2018 Mallhi et al.
Copyright year: 2018
Copyright holder: Mallhi et al.
License: This is an open access article distributed under the terms of the Creative Commons Attribution License, which permits unrestricted use, distribution, reproduction and adaptation in any medium and for any purpose provided that it is properly attributed. For attribution, the original author(s), title, publication source (PeerJ) and either DOI or URL of the article must be cited.
License URL: https://creativecommons.org/licenses/by/4.0/

Keywords: Chikungunya, Epidemic, Vector borne diseases, Pakistan, Outbreak, Viral infection

Funding: The authors received no funding for this work.

==============================
Background

The World Health Organization (WHO) declares Chikungunya (CHIK) infection to be endemic in South Asia. Despite its first outbreak in Pakistan, no documented evidence exists which reveals the knowledge or awareness of healthcare students and workers (HCSW) regarding CHIK, its spread, symptoms, treatment and prevention. Since CHIK is an emergent infection in Pakistan, poor disease knowledge may result in a significant delay in diagnosis and treatment. The current study was aimed to evaluate the awareness and knowledge of CHIK among HCSW.

Methods

A cross-sectional study was conducted among HCSW from teaching institutes and hospitals in seven provinces of Pakistan. We collected information on socio-demographic characteristics of the participants and their knowledge by using a 30-item questionnaire. The cumulative knowledge score (CKS) was calculated by correct answers with maximum score of 22. The relationship between demographics and knowledge score was evaluated by using appropriate statistical methods.

Results

There were 563 respondents; mean age 25.2 ± 5.9 years with female preponderance (62.5%). Of these, 319 (56.7%) were aware of CHIK infection before administering the survey. The average knowledge score was 12.8 ± 4.1 (% knowledge score: 58.2%). Only 31% respondents had good disease knowledge while others had fair (36.4%) and poor (32.6%) knowledge. Out of five knowledge domains, domain III (vector, disease spread and transmission) and V (prevention and treatment) scored lowest among all i.e. percent score 44.5% and 54.1%, respectively. We found that socio-demographic characteristics had no influence on knowledge score of the study participants.

Conclusion

Approximately one-half of participants were not aware of CHIK infection and those who were aware had insufficient disease knowledge. Findings of the current study underscore the dire need of educational interventions not only for health care workers but also for students, irrespective to the discipline of study.

Introduction

Chikungunya virus (CHIKV; genus Alphavirus, family Togaviridae) is transmitted to humans by Aedes mosquitos in sylvatic (animal-mosquito-man) or urban (man-mosquito-man) transmission cycle and was first identified in Tanzania in the 1950s (Costa-da Silva et al., 2017). Since its introduction to the New World, Chikungunya (CHIK) has caused impressive outbreaks in Africa, Europe, Asia, and islands of the Indian and western Pacific Oceans (Pastula et al., 2017). According to a recent estimate, 3.6 billion people living in 124 countries are at high risk of disease with infection rates reported up to 75% (Nasci, 2014). In summer 2017, Pakistan experienced its first ever CHIK outbreak in Karachi, a metropolis of approximately 25 million inhabitants. This outbreak resulted in 30,000 suspected cases where only 803 were reported to World Health Organization (WHO) (Mallhi et al., 2017a). Shi et al. (2017) carried out phylogenetic analysis of CHIKV isolated from the 10 patients with confirmed diagnosis and found that the epidemic of genotypes of CHIKV strains were tightly associated with spatial and temporal distributions. This analysis revealed that Pakistani strains shared high similarity and belonged to ECSA.IOL lineage. Moreover, authors found that the strains were closely related to those derived in India which suggests the possibility of their migration from India to Pakistan. Naqvi et al. (2017) evaluated clinico-laboratory spectrum of 199 isolated CHIK cases from the emergency departments of four tertiary care hospitals in Karachi. The most common clinical manifestations among these patients were lymphocytosis, joint pain and swelling (in small joints) followed by high grade fever (>102°F), myalgia and thrombocytopenia. The duration of illness was greater than 30 days in 62 patients which was attributed to the persistent joint pain. Moreover authors also interviewed patients about the etiological factors where most of them responded mosquitos as causative agents followed by chickens. The evidence of CHIKV in Pakistan dates back to the early 1980s, when Darwish et al. (1983) identified antibodies against CHIKV in sera of rodents and humans. Fortunately, no hospital case was reported on that occasion. Later, three children with CHIK infection were identified during a dengue outbreak in 2011 (Afzal et al., 2015).

The year 2016 portrayed a disturbing picture, when Pakistan experienced a noticeable burden of viral infections, including 19 deaths caused by Crimean-Congo haemorrhagic fever (CCHF) and outbreaks of dengue and CHIK. The trail of viral attacks has created many concerns among health authorities and WHO (Mallhi et al., 2017b). In recent times, with an increase in global travel, the risk for spreading CHIK to non-endemic regions has heightened. Like dengue epidemics, CHIK outbreak is a result of abrupt global expression of vector-borne diseases (Charrel, De Lamballerie & Raoult, 2007). CHIK is self-limiting and has a low mortality rate; however, fatal infections and chronic rheumatic disorders do occur (Charrel, De Lamballerie & Raoult, 2007). The infection not only poses adverse impact on human health but also contributes to overall socioeconomic burden of the community and health care system (Cardona-Ospina, Villamil-Gómez & Jimenez-Canizales, 2015). The virus can establish itself in any tropical or temperate region harboring the Aedes mosquitoes. Thus, the key measures for preventing CHIK epidemics include entomologic surveillance, peridomestic mosquito control, public education, commitment of resources for research, improvements in healthcare infrastructure, detection of imported cases and early recognition of local transmission, followed by efficient vector control (Mallhi et al., 2017a; Mallhi et al., 2017b).

Given the absence of licensed vaccine and specific drug treatments for CHIK infection, the Centers for Disease Control and Prevention (CDC) has focused on, among other things, raising awareness for both health care providers and general public (CDC, 2015). Though most intervention strategies have focused on mosquito control and mosquito bite prevention, the success of these strategies relies on social factors such as knowledge and awareness of diseases. Healthcare professionals (HCPs) serve as the first-line of CHIK diagnosis, notification, and treatment and poor disease knowledge may result in significant delay in a patient’s detection and management which may further be associated with the spread of disease. HCPs also play a pivotal role in providing education, increasing public awareness and promoting personal protection. In this context, knowledge and awareness of disease among HCPs must be appropriate and up-to-date, which could be translated into early recognition and improved outcomes of CHIK control. Although WHO and CDC have embarked on a CHIK awareness campaign for HCPs, the level and extent of awareness remain unknown (Omodior et al., 2017). There is a dearth of investigations on awareness and knowledge of CHIK infection among healthcare students and workers (HCSW) in Pakistan. Furthermore, it is imperative to explore knowledge of CHIK among HCSW after its outbreak in Pakistan in order to evaluate their preparedness for the re-emergence of the virus. In this context, the current study was aimed to evaluated awareness and knowledge of CHIK infection among HCSW of Pakistan.

Methods

Ethics statement

The current study was approved by the Human Research Ethics Committee (HREC) at Government College University Faisalabad, Pakistan (HREC/Phar/GCUF/2017-332). Informed consent was obtained from all participants and data were anonymised before analysis.

Study site and population

This cross-sectional study (July 2017 to December 2017) was conducted among HCSW from all seven provinces or administrative states of Pakistan. We selected five major categories of health professions including pharmacists, physicians, dentists, physiotherapists and nurses. The study flow diagram is presented in Fig. 1.

Figure 1 Study flow diagram.

Study instrument

A 30-items questionnaire comprised of three sections was developed under opinions of experts from five health professions (doctor, pharmacist, nurse, dentist, and physiotherapist). Upon completion of content validity, the survey instrument was pre-tested in a small, targeted sample of HCSW (n = 30), with the aim of assessing the clarity and comprehensibility of questions (face validity). The reliability scale was applied for these 30 respondents and the alpha value was found at 0.811, indicating the adequacy of the tool to meet the objectives of the current study. Each section of the questionnaire included close-ended questions. ‘Introduction’ was comprised of six items of demographics, while ‘Methods’ consisted of two questions evaluating the general awareness of infection. The participants, who did not hear the word “chikungunya” before administering the survey were considered as “not aware of disease” and were not included in the knowledge score analysis in ‘Results’. ‘Results’ was comprised of 22 items which were categorized in five domains. These domains evaluated the knowledge of study participants regarding CHIK infection such as: knowledge of recent outbreak in Pakistan (two items), basic disease knowledge (two items), knowledge of vectors, disease spread and transmission (six items), symptomology (nine items), and prevention and treatment of CHIK infection (three items). The knowledge of participants was scored “1” for each correct answer and scored “0” for incorrect or don’t know or not sure answers. The maximum cumulative knowledge score (CKS) was 22. Total percent knowledge score (Score obtained/CKS ×100) was calculated for each participant and knowledge of CHIK was categorized into good (score ≥ 70), fair (score 50.1–69.9) and poor (score ≤ 50). The percent knowledge score was also calculated against each domain of ‘Results’. A similar scoring system has been previously adopted in several investigations (Khan et al., 2014; Khan, Sarriff & Khan, 2012).

Data collection

Using convenient sampling technique, all the authors were asked to contact HCSW from universities, hospitals and community health centers in each province of Pakistan for interview. Authors explained the purpose of study to the target population and those who agreed to participate were asked to fill the questionnaire. An informed consent was obtained from each participant. Each questionnaire was collected and all the participants were educated on CHIK infection, its prevention and treatments. At the end of study period, all the responses were checked for completeness and data were transferred to a Microsoft spreadsheet for cleaning purposes.

Statistical analysis

All the data were analyzed by SPSS version 22.0. A significance level of 0.05 was used throughout. All continuous variables were reported as mean (standard deviation) or median (25%–75% IQR), while categorical variables were described using counts (n) and proportions (%). Chi-square test or student-t test was used to compare the demographics between respondents who were aware of CHIK and those who were not. Association between knowledge score and socio-demographic variables was evaluated by simple linear regression analysis, Pearson correlation or one-way ANOVA, where appropriate.

Results

Out of 814, a total of 618 questionnaires were received (response rate: 76%), and 563 responses were included for the analysis after excluding 55 uncompleted forms (Fig. 1). The mean age of the participants were 25.2 ± 5.9 years with female preponderance (62.5%). Most of the respondents were students (58.8%), while 36% were working professionals. About half of the responses were recorded from the pharmacy profession, followed by doctors (34.1%) and dentists (8%). Four-hundred (71%) participants were at graduation level. Being a post populated province of Pakistan, the majority of the responses (73.9%) were recorded from Punjab. Table 1 demonstrates the general demographics of the study participants and compares the respondents who were aware of CHIK infection with those who were not.

Table 1 Demographics of study participants and their association with chikungunya awareness.

	Total participantsN = 563	Participants who were not aware of CHIK (unawareness)N = 244	Participants who were aware of CHIK (awareness)N = 319	P*value	
Age (years)	25.2 ± 5.9	24.7 ± 7.1	25.6 ± 4.9	0.066	
18–25 Years	375 (66.6%)	181 (74.2%)	194 (60.8%)	0.001	
26–39 Years	174 (30.9%)	57 (23.4%)	117 (36.7%)	0.001	
≥40 Years	14 (2.5%)	6 (2.5%)	8 (2.5%)	0.971	
Gender				0.799	
Male	211 (37.5%)	90 (36.9%)	121 (37.9%)		
Female	352 (62.5%)	154 (63.1%)	198 (62.1%)		
Working status					
Student	331 (58.8%)	161 (66%)	170 (53.3%)	0.002	
Working	205 (36.4%)	69 (28.3%)	136 (42.6%)	0.001	
Unemployed	27 (4.8%)	14 (5.7%)	13 (4.1%)	0.360	
Field of education				0.002	
Pharmacy	282 (50.1%)	103 (42.2%)	179 (56.1%)	0.001	
MBBS	192 (34.1%)	77 (31.6%)	115 (36.1%)	0.265	
BDS	45 (8%)	31 (12.7%)	14 (4.4%)	<0.001	
Physiotherapy	19 (3.4%)	14 (5.7%)	5 (1.6%)	0.007	
Nursing	25 (4.4%)	19 (7.8%)	6 (1.9%)	0.001	
Level of education				0.003	
Graduation	400 (71%)	189 (77.5%)	211 (66.1)		
Post-graduation	163 (29%)	55 (22.5%)	108 (33.9)		
Provinces				<0.001	
Punjab	416 (73.9%)	187 (76.6%)	229 (71.8%)	0.191	
Sindh	39 (6.9%)	3 (1.2%)	36 (11.3%)	<0.001	
Balochistan	14 (2.5%)	7 (2.9%)	7 (2.2%)	0.611	
Khyber Pakhtunkhwa (KPK)	73 (13%)	34 (13.9%)	39 (12.2%)	0.550	
Federally Administered tribal areas (FATA)	13 (2.3%)	8 (3.3%)	5 (1.6%)	0.180	
Gilgit Baltistan (GB)	3 (0.5%)	1 (0.4%)	2 (0.6%)	0.726	
Azad Jammu & Kashmir (AJK)	5 (0.9%)	4 (1.6%)	1 (0.3%)	0.97	
Notes.

MBBS Bachelor of Medicine, Bachelor of Surgery, BDS: Bachelor of Dental Surgery

* P values represents the difference of demographics between awareness and unawareness of chikungunya, calculate by Chi-square Test or Fischer Exact Test for categorical variables and student-t test for continuous variable (age).

Alarmingly, 244 (43.3%) respondents had never heard about CHIK infection before administering the survey, while 319 (56.7%) participants stated that they were aware of the disease. Among respondents who were aware, the primary source of information was television (33.5%) followed by social networks (25.08%) and newspapers (15.05%) (Fig. 2). The association of awareness with demographics was evaluated using Chi-square statistics (Table 1). The respondents with age 26–39 years had CHIK awareness (p = 0.001), while those of 18–25 years were unaware of the disease (p = 0.001). Gender had no impact on disease awareness in the present study. Health students were significantly (p = 0.002) associated with unawareness of disease, while health professionals were aware of CHIK (p = 0.001). Among health professions included in the current study, pharmacy and MBBS were associated with disease awareness, while all other professions were statistically associated with unawareness. Only the respondents from Sindh province had significant awareness of CHIK infection.

Figure 2 Sources of information of Chikungunya infection among study participants (n = 319).

The mean cumulative knowledge score (CKS) was 12.8 ± 4.1 (Median: 14, range” 0–21, percent knowledge score: 58.2%). The proportion of participants with good knowledge was 31%, while 36.4% and 32.6% respondents had fair and poor knowledge of CHIK, respectively. The least scored (44.5%) domain was “knowledge of vector, disease spread and transmission” followed by the domain “knowledge of prevention and treatment” (54.1%). However, the basic disease knowledge of the respondents was good (% score: 71.8%). The percentage score ranged from fair to poor in all other domains assessing the knowledge among participants. Table 2 demonstrates questions evaluating the knowledge of participants with the average knowledge score of each item and % knowledge score of each domain.

Table 2 Knowledge questions and their responses by the participants with average knowledge score (AKS) of each item.

Questions	Responses (N = 319)	AKS	
Domain I: Knowledge of recent outbreak in Pakistan [% Score (obtained score/total score): 61.4%]	1.23 ± 0.84	
1. Do you know that Chikungunya outbreak has been reported in Pakistan?		0.73 ± 0.44	
A. Yes✓ 	233 (73%)		
B. No	86 (27%)		
2. If above answer is “Yes” then where has the Chikungunya outbreak recently occurred?		0.50 ± 0.50	
A. Lahore	24 (7.5%)		
B. Karachi✓ 	140 (43.9%)		
C. Multan	–		
D. Faisalabad	3 (0.9%)		
E. Islamabad	–		
F. Not sure	66 (20.7%)		
G. Did not respond	86 (27%)		
Domain II: Basic disease knowledge [% Score (obtained score/total score): 71.8%]	1.44 ± 0.75	
3. Chikungunya is a		0.77 ± 0.42	
A. Bacterial Infection	17 (5.3%)		
B. Viral Infection✓ 	247 (77.4%)		
C. Not sure	55 (17.2%)		
4. Which infection is closely related to Chikungunya infection?		0.66 ± 0.47	
A. Pneumonia	3 (0.9%)		
B. Dengue Infection✓ 	211 (66.1%)		
C. Ebola Infection	30 (9.4%)		
D. Not sure	75 (23.5%)		
Domain III: Knowledge of vector, disease spread & transmission [% Score (obtained score/total score): 44.5%]	2.67 ± 1.31	
5. Chikungunya is caused by mosquito bite, what is the name of the mosquito?		0.45 ± 0.50	
A. Anopheles	18 (5.6%)		
B. Adese✓ 	144 (45.1%)		
C. Both	23 (7.2%)		
D. Not sure	134 (42%)		
6. What is the common breeding site of Chikungunya mosquito?		0.49 ± 0.50	
A. Water storage containers/Stagnant water✓ 	156 (48.2%)		
B. Dirty water	36 (11.3%)		
C. Garbage and mud	26 (8.2%)		
D. Not sure	101 (31.7%)		
7. During which time do Chikungunya mosquitos bite preferably?		0.59 ± 0.49	
A. Day	75 (23.5%)		
B. Night	56 (17.6%)		
C. Anytime✓ 	187 (58.6%)		
D. Not sure	1 (0.3%)		
8. During which season is Chikungunya infection most common?		0.55 ± 0.50	
A. Dry summer	75 (23.5%)		
B. Monsoon✓ 	176 (55.2%)		
C. Winter	18 (5.6%)		
D. Spring	24 (7.5%)		
E. Not sure	26 (8.2%)		
9. Does Chikungunya infection transfer from direct human to human contact?		0.34 ± 0.47	
A. Yes	69 (21.6%)		
B. No✓ 	108 (33.9%)		
C. Not sure	142 (44.5%)		
10. Does Chikungunya infection transfer from mother to new born child?		0.25 ± 0.43	
A. Yes✓ 	81 (25.4%)		
B. No	53 (16.6%)		
C. Not sure	185 (58%)		
Domain IV: Symptomology [% Score (obtained score/total score): 64.5%]	5.81 ± 2.43	
11. Is fever a symptom of Chikungunya?		0.89 ± 0.32	
A. Yes✓ 	283 (0.6%)		
B. No	2 (88.7%)		
C. Not sure	34 (10.7%)		
12. Is joint pain a symptom of Chikungunya?		0.77 ± 0.42	
A. Yes✓ 	247 (77.4%)		
B. No	11 (3.4%)		
C. Not sure	61 (19.1%)		
13. Is muscle pain a symptom of Chikungunya?		0.72 ± 0.45	
A. Yes✓ 	231 (72.4%)		
B. No	7 (2.2%)		
C. Not sure	81 (25.4%)		
14. Is headache a symptom of Chikungunya?		0.71 ± 0.45	
A. Yes✓ 	228 (71.5%)		
B. No	11 (3.5%)		
C. Not sure	80 (25.1%)		
15. Is nausea a symptom of Chikungunya?		0.48 ± 0.50	
A. Yes✓ 	145 (48.3%)		
B. No	37 (11.6%)		
C. Not sure	128 (40.1%)		
16. Is fatigue a symptom of Chikungunya?		0.75 ± 0.44	
A. Yes✓ 	238 (74.6%)		
B. No	13 (4.1%)		
C. Not sure	68 (21.3%)		
17. Is fever a symptom of Chikungunya?		0.56 ± 0.50	
A. Yes✓ 	179 (56.1%)		
B. No	28 (8.8%)		
C. Not sure	112 (35.1%)		
18. After the bite of infected mosquito, how many days does it take for symptoms to appear?		0.54 ± 0.50	
A. Abruptly (immediately after bite)	11 (3.4%)		
B. 3–7 days✓ 	173 (54.2%)		
C. On next day of mosquito bite	11 (3.4%)		
D. Not sure	124 (38.9%)		
19. For how many days do symptoms of Chikungunya last?		0.38 ± 0.49	
A. One month	48 (15%)		
B. 7–10 days✓ 	120 (37.6%)		
C. One day	2 (0.6%)		
D. Not sure	49 (46.7%)		
Domain V: Prevention and treatment [% Score (obtained score/total score): 54.1%]	1.62 ± 1.02	
20. Is Chikungunya a preventable disease?		0.78 ± 0.42	
A. Yes✓ 	249 (78.1%)		
B. No	12 (3.8%)		
C. Not sure	58 (18.2%)		
21. Is there any specific drug available for Chikungunya treatment?		0.50 ± 0.50	
A. Yes	38 (11.9%)		
B. No✓ 	159 (49.8%)		
C. Not sure	122 (38.2%)		
22. Is there any vaccine available for Chikungunya prevention?		0.34 ± 0.48	
A. Yes	61 (19.1%)		
B. No✓ 	110 (34.5%)		
C. Not sure	148 (46.4%)		
	
Notes.

✓ Represents the correct answer.

AKS, average knowledge score with standard deviation.

The knowledge score was equally distributed between demographics and we found no relationship of demographic parameters with knowledge score. Though the knowledge score was higher in age 26–39 years, working professionals, MBBS field of education and Sindh province, statistical significant association was not present as compared to others (Table 3).

Table 3 Distribution of knowledge score among demographics of study participants.

	RespondentsN = 319	Knowledge score (out of 22)	P∗ value	
Age (years)	25.6 ± 4.9		r: 0.06 (P = 0.274)a	
Age Categories			0.798b	
18–25 Years	194 (60.8%)	12.7 ± 4.4		
26–39 Years	117 (36.7%)	13.0 ± 5.0		
≥ 40 Years	8 (2.5%)	12.4 ± 6.0		
Gender			0.709c	
Male	121 (37.9%)	12.9 ± 4.7		
Female	198 (62.1%)	12.7 ± 4.7		
Working status			0.104b	
Student	170 (53.3%)	12.4 ± 4.7		
Working	136 (42.6%)	13.4 ± 4.4		
Unemployed	13 (4.1%)	11.3 ± 6.3		
Field of education			0.486b	
Pharmacy	179 (56.1%)	12.7 ± 4.8		
MBBS	115 (36.1%)	13.1 ± 4.6		
BDS	14 (4.4%)	12.6 ± 3.2		
Physiotherapy	5 (1.6%)	9.2 ± 5.5		
Nursing	6 (1.9%)	12.2 ± 3.1		
Level of education			0.924c	
Graduation	211 (66.1%)	12.8 ± 4.4		
Post-graduation	108 (33.5%)	12.7 ± 5.1		
Provinces			0.648b	
Punjab	229 (71.8%)	12.8 ± 4.8		
Sindh	36 (11.3%)	13.6 ± 3.4		
Balochistan	7 (2.2%)	10.6 ± 5.5		
Khyber Pakhtunkhwa (KPK)	39 (12.2%)	12.3 ± 4.7		
Federally Administered tribal areas (FATA)	5 (1.6%)	10.8 ± 2.1		
Gilgit Baltistan (GB)	2 (0.6%)	14.5 ± 2.1		
Azad Jammu & Kashmir (AJK)	1 (0.3%)	12.0		
Notes.

a Pearson correlation.

b One-Way ANOVA.

c Student t-test.

The relationship between demographics and CKS was further assessed with simple linear regression (Table 4). Though demographics made a contribution to explaining the dependent variable (i.e., CKS), the association was statistically insignificant.

Table 4 Simple linear regression analysis examining the contribution of demographics to cumulative knowledge score (CKS).

Variables	Beta	Sig.	95% Confidence interval	
	Unstandardized	Standardized		Lower bound	Upper bound	
Age	0.303	0.032	0.571	0.748	1.353	
Gender	−0.201	−0.021	0.709	−1.257	0.856	
Working status	0.090	0.101	0.172	0.939	1.119	
Field of education	−0.417	−0.043	0.442	−1.484	0.657	
Level of education	−0.055	−0.006	0.920	−1.139	1.028	
Provinces	−0.919	−0.063	0.265	−2.337	0.699	
Notes.

Simple linear correlation (dependent variable: cumulative knowledge score, independent variables: binary demographic variables).

Reference (constant): age (≤ 25 years), gender (male), working status (student), field of education (MBBS), level of education (graduation), provinces (Sindh).

Discussion

To the best of our knowledge, this is the first nationwide study to explore the awareness and extent of knowledge about basics, spread, transmission, symptoms, prevention and treatment of CHIK infection among healthcare students and workers (HCSW) after its outbreak in Pakistan. Previous closely related investigations conducted in Pakistan either included very few participants or were limited to HCSW from one or two cities (Gul, Aziz & Tarik, 2014; Mansoor et al., 2017).

The main findings of the present study revealed that HCSW had inadequate awareness and knowledge of CHIK infection. It is pertinent to mention that approximately half of the study participants did not hear the word “chikungunya” before administering the survey. Another recent investigation in Pakistan reported that 18.8% of healthcare professionals had never heard of the disease (Mansoor et al., 2017). The level of awareness reported by Mansoor et al. (2017) was higher as compared to our findings, which might be attributed to the inclusion of only physicians and study location, as all the study participants belonged to well-equipped and reputed hospitals located in the capital of Pakistan. Authors have also described that CHIK infection is not an important part of the syllabi of physicians. Being HCSW, provision of adequate information and education to the public is an ethical obligation. However, to carry out such services, their disease knowledge must be sufficient and up-to-date. Clinical and epidemiological similarities with dengue fever make CHIK diagnosis difficult, which may lead physicians to misdiagnose CHIK as dengue fever, particularly in parts of world where awareness of CHIK is scarce. Pakistan is one such country in which a CHIK outbreak was initially reported as “mysterious disease” (Dunya, 2016) and it might be attributed to the inadequate awareness of disease among healthcare professionals. A previous survey conducted on HCSW in Pakistan indicated that about 25% of the study population had very little information of CHIK (Gul, Aziz & Tarik, 2014). Lack of awareness in large proportions of HCSW in the present study is alarming. Health and teaching institutes should arrange talks or awareness programs immediately after reporting of disease epidemic, in order to ensure effective preparedness for future events. Our findings indicate that healthcare students from pharmacy, dentistry, physiotherapy and nursing, who were at graduation level of education from all provinces, except Sindh, were associated with unawareness of disease. However, the awareness was comparatively higher among working professionals, especially in Sindh Province. It might be attributed to the reason that the first CHIK epidemic occurred in Sindh Province of Pakistan. Most of the study participants reported that they had heard about CHIK through television or electronic media. Similar findings have been indicated by Mansoor et al. (2017) where 43% physicians reported electronic media as a primary source of disease awareness. In addition to the electronic media and social networks, health authorities must ensure disease awareness campaigns in hospitals and health teaching institutes.

Knowledge scoring was done among participants who were aware of Chikungunya (N = 319) by excluding the respondents who had never heard about the disease. Though electronic media played a pivotal role in highlighting the emergence of the disease in Pakistan (Dunya, 2016) still one fourth of the participants had no knowledge of the recent outbreak of CHIK in the country. Approximately, 77% of respondents were aware of the etiological cause of the disease and these findings are consistent with the results of a previous investigation (Mansoor et al., 2017). In contrast, Gul, Aziz & Tarik (2014) reported that only 22% of healthcare professionals were aware of the disease cause. It is important to mention that the knowledge score regarding disease vector, spread and transmission was lowest among all domains. This domain describes the vector control measures along with disease teratogenicity and such a low score depicts that HCSW are not well prepared to educate the general public. These findings instigate the dire need of disease education, especially concerning emergent infectious diseases, among HCSW.

The average score of symptomology domain was fair (64.5%) in the current study and is comparable with the study conducted in Pakistan where 65.5% participants reported correct symptoms of CHIK (Mansoor et al., 2017). Another study investigating the knowledge of CHIK among HCSW in Colombia reported that 92% respondents correctly reported the symptoms of disease (Bedoya-Arias et al., 2015). Fever was answered as a frequent presentation of CHIK while nausea and rash were least scored by the participants in the present study. Since CHIK shares common symptoms with dengue and Zika, appropriate knowledge on disease presentations is a cardinal feature to distinguish these closely related infectious diseases and to ensure the correct and timely diagnosis. Accurate knowledge of disease manifestations is a mainstay of referral or successful therapy. Clinical and epidemiological similarities with dengue fever make CHIK diagnosis difficult, which may lead physicians to misdiagnose CHIK as dengue fever; therefore, the incidence of CHIK may actually be higher than currently believed (Thiboutot et al., 2010). We urge regulatory authorities to ensure the continuous medical education (CME) for health professionals, specifically for diseases requiring differential or distinctive diagnoses.

Currently, CHIK is treated symptomatically, usually with non-steroidal anti-inflammatory drugs or steroids, bed rest, and fluids (Thiboutot et al., 2010). Alarmingly, more than half of the study participants were not aware of CHIK treatment, where 38% respondents reported that they were not sure whether any specific medication is available for its treatment. Despite the gravity of its infectious potency and the fear of it being a potential biological weapon, there is currently no vaccine for CHIK infections. Only 34% of HCSW responded that there is no commercial vaccine for CHIK while 12% agreed to its availability and 38% were not sure. Low knowledge score of prevention and treatment domain among HCSW warrants the earnest maneuvers by the health authorities. The relationship between demographics and knowledge score was evaluated among studied participants with no influence of demographic characteristics on knowledge scoring. These findings insist the need of health professional education initiatives for epidemic diseases throughout all disciplines of health care system.

A crucial element in vector-borne diseases is behavioral change. WHO works with partners to provide education and improve awareness so that people know how to protect themselves and their communities from mosquitoes, ticks, bugs, flies and other vectors (WHO, 2017). An effective public health education can only be possible with appropriate disease knowledge among HCSW. To ensure all graduating health professionals are prepared to engage in public health activities, education in this field must be provided during their main years of education (Law et al., 2017). Public health education should be incorporated into the curricula of health professional studies within developing nations so all graduates are prepared to engage in public health activities.

Performing studies focused on quantifying and reinforcing knowledge among HCSW in regions with high prevalence of CHIK is important for disease outbreak preparedness. Knowledge about the vector involved in its transmission plays an important role in disease prevention. Finally, symptomatology recognition by the community leads to timely admission to health centers for optimal disease management (Bedoya-Arias et al., 2015).

The findings of this study are, however, limited to the HCSW of Pakistan represented in the sample and cannot be generalized to the broader context. However, the findings can be implicated to all four provinces of Pakistan to initiate targeted measures. It is quite possible that the segment of the population not represented in the study (senior skilled professionals who refused to participate due to their busy schedule) is significantly different in some respect. Moreover, the participation from small administrative states including FATA, GB and AJK was limited which precludes the implications of the findings in these states. The proportion of responses from dentists, physiotherapists and nurses was comparatively less which may bias the findings towards physicians and pharmacists and underscore the consideration of equal response proportion for future studies. The knowledge evaluation in this study was based on very basic questions and there is high propensity that detailed knowledge analysis may yield poor scores among participants. Nevertheless, the current study is strengthened by the first nationwide survey including a large pool of HCSW from various disciplines and evaluates the extent of knowledge and their relationship with demographic profile. The findings of the present study will serve to design and implement disease knowledge initiatives by the health authorities in the existing vector control programs.

Conclusions

The awareness and knowledge of CHIK infection among healthcare students and workers are insufficient to meet the standards of preparedness for future outbreak events. Since Pakistan is experiencing the quadruple burden of vector borne diseases (VBDs), adequate information on these diseases among HCSW and their effective provision to the general community is essential to quell the growing risks of disease spillover. Our findings underscore the need of multidimensional approaches to educate HCSW for emerging VBDs. More comprehensive and elaborated nationwide surveys are needed to strategize the targeted education plan for health professionals.

Supplemental Information

Supplemental Information 1 Study tool

Click here for additional data file.

Supplemental Information 2 SPSS file

Each data point indicates the individual responses of participants to each question asked during data collection.

Click here for additional data file.

Supplemental Information 3 SPSS file of participants who were aware of Chikungunya

Each data point indicates the responses of participants who were aware of Chikungunya infection.

Click here for additional data file.

Additional Information and Declarations

Competing Interests

Author Contributions

Human Ethics

Data Availability

The authors declare there are no competing interests.

Tauqeer Hussain Mallhi conceived and designed the experiments, performed the experiments, analyzed the data, contributed reagents/materials/analysis tools, authored or reviewed drafts of the paper, approved the final draft.

Yusra Habib Khan conceived and designed the experiments, performed the experiments, analyzed the data, authored or reviewed drafts of the paper, approved the final draft, performed the statistical analysis.

Nida Tanveer performed the experiments, contributed reagents/materials/analysis tools.

Allah Bukhsh performed the experiments, prepared figures and/or tables, authored or reviewed drafts of the paper.

Amer Hayat Khan performed the experiments, analyzed the data, approved the final draft, performed the statistical analysis.

Raja Ahsan Aftab and Omaid Hayat Khan performed the experiments, prepared figures and/or tables.

Tahir Mehmood Khan performed the experiments, contributed reagents/materials/analysis tools, authored or reviewed drafts of the paper, approved the final draft.

The following information was supplied relating to ethical approvals (i.e., approving body and any reference numbers):

The Human Research Ethics Committee (HREC) at Government College University Faisalabad approved this study (HREC/Phar/GCUF/2017-332).

The following information was supplied regarding data availability:

The raw data are provided in the Supplemental Files.

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
