# Peer review of "Awareness and knowledge of Chikungunya infection following its outbreak in Pakistan among health care students and professionals: a nationwide survey"

_PeerJ, doi:10.7717/peerj.5481_

## Round 0.1 · original submission · Major Revisions

· Academic Editor

Major Revisions

Your manuscript has been reviewed by four experts and they have raised issues of self-plagiarism, and questioned the methodology and lack of review of relevant literature. These will need to be addressed in a revised manuscript. If you do submit a revised manuscript, please also have it checked for English language use, as PeerJ does not offer copy-editing.

Reviewer 1 ·

Basic reporting

.

Experimental design

.

Validity of the findings

.

Additional comments

Awareness and knowledge of Chikungunya infection following its outbreak in Pakistan among health care students and professionals: A nationwide survey
(#25826) by Mallhi et al.,

General comments
The authors is addressing interesting research question of Chikungunya Awareness among Healthcare workers and students which currently is a global health problem.

The manuscript has been well written and all the analysis done appropriately.
However, there are few comments.


Data Collection
Line 127, page 12,…….all authors were asked to contact HCSW!. Did all authors participated in data collection?

Line 135, page 12
Please provide the manufacturers name.
Figure 1
The author should explain why figure 1 is important!. Text explanation is enough.


Line 196-197
The level of awareness reported by Mansoor et al. was higher as compared to our
findings, which might be ……
How did you measure level of awareness?
How much did you got?
Awareness and Knowledge do they represent the same thing in this study?

Reviewer 2 ·

Basic reporting

no comment

Experimental design

Add in methodology the criteria to measure and evaluate awareness about the infection

Validity of the findings

no comment

Additional comments

Line 45 correct the words Adese mosquitoes by Aedes mosquitoes

Reviewer 3 ·

Basic reporting

The abstract should be reviewed and edited by a native English speaker as it is difficult to read and understand at this stage.

There is a lack of information about the epidemiological context. Was the magnitude of the epidemic estimated? Were there any prevalence survey conducted in the country?

There is no review of literature about the knowledge of chikungunya in the region prior to the discussion. As much as I know, several KABP studies were conducted in the Indian Ocean in the late 2000ies.

Experimental design

How were the participants slected and recruited? There is no information about this...

Validity of the findings

Conclusion are well stated, and linked to original research question.

Reviewer 4 ·

Basic reporting

The authors Mallhi et al. have presented their findings on Awareness and Knowledge of chikungunya infection in health care students and workers in Pakistan. The text needs significant improvement and clarification and must not repeat sentences from already published articles.

For instance: the authors have copied text from an already published manuscript.
"First chikungunya outbreak in Pakistan: a trail of viral attacks"
New Microbes New Infect. 2017 Sep; 19: 13–14.
Published online 2017 Jun 1.
doi: 10.1016/j.nmni.2017.05.008.

In the Introduction, 1st Paragraph:
"The history of chikungunya in Pakistan dates to the early 1980s, when Darwish et al. [1] in 1983 detected antibodies against the chikungunya virus from sera of four rodents and one human. However, virus spillover did not occur on that occasion. Later, in 2015, three cases of chikungunya were identified in children during a 2011 dengue outbreak"

2nd paragraph:
"Pakistan has experienced a quadruple burden of viral infections in 2016, including Crimean-Congo haemorrhagic fever and outbreaks of dengue and chikungunya. These three consecutive viral infections have created much concern among health authorities and the World Health Organization".

The above mentioned statements are exactly copied from an article on Chikungunya published by authors, with some of them sharing both manuscripts. As a rule, even the manuscript on the similar subject by the same authors should not be replicated as such.

Experimental design

The authors have claimed to present their findings based on a "nationwide" survey. However, the data is biased with over-representation from a province "Punjab" with 229 respondents, compared to only a single respondent from "Azad Jammu & Kashmir (AJK) and less than ten in each of region FATA, GB, Balochistan. This ambiguity poses a serious challenge to target areas where the disease control interventions need to be initiated.

Validity of the findings

The authors should analyze data based on the hospitals known to receive and treat chikungunya patients compared to those without attending such patients. Without this, the true assessment on knowledge and awareness may not be revealed effectively.

The authors should work on data uniformity across all study groups, for example, among the 319 respondents, there were 179 (56%) pharmacist and the total number of nurses, who usually are at much higher risk, were just 6 (2%).

Additional comments

The authors should significantly improve the draft as per comments given in above three sections: Basic reporting, Experimental Design and Validity of Findings. The manuscript must be reviewed by an expert in English language.

---

## Round 0.2 · Minor Revisions

· Academic Editor

Minor Revisions

Please make the remaining minor revisions requested by the reviewer. Please also ensure the formatting of the references is correct.

Reviewer 4 ·

Basic reporting

A brief on the epidemiology of recent outbreak of chikungunya in Pakistan can be added to the Introduction section, quoting the two recently published articles:

https://www.ncbi.nlm.nih.gov/pmc/articles/PMC5587410/pdf/cureus-0009-00000001430.pdf
https://link.springer.com/content/pdf/10.1007%2Fs12250-017-4077-5.pdf

Experimental design

No Comment

Validity of the findings

No Comment

Additional comments

The authors have significantly improved the manuscript draft.

Major Revision:
The Introduction section should be updated by quoting recent outbreak event of chikungunya in Pakistan as presented in the below-mentioned two articles:

https://www.ncbi.nlm.nih.gov/pmc/articles/PMC5587410/pdf/cureus-0009-00000001430.pdf
https://link.springer.com/content/pdf/10.1007%2Fs12250-017-4077-5.pdf

Minor Revisions:

The format of references should be corrected and unified as per journal's standards.

---

## Round 0.3 · accepted · Accept

· Academic Editor

Accept

Thanks for dealing with those last minor corrections. We look forward to seeing your work in press!